# Effects of Laser Defocusing on Bead Geometry in Coaxial Titanium Wire-Based Laser Metal Deposition

**DOI:** 10.3390/ma17040889

**Published:** 2024-02-15

**Authors:** Remy Mathenia, Aaron Flood, Braden McLain, Todd Sparks, Frank Liou

**Affiliations:** 1Department of Mechanical and Aerospace Engineering, Missouri University of Science and Technology, Rolla, MO 65409, USA; remymathenia@mst.edu (R.M.); btmywv@mst.edu (B.M.); 2Product Innovation and Engineering, St. James, MO 65559, USA; ajflood@mopine.com (A.F.); toddesparks@mopine.com (T.S.)

**Keywords:** three-beam coaxial laser, wire deposition, bead geometry, defocusing

## Abstract

Coaxial wire-based laser metal deposition is a versatile and efficient additive process that can achieve a high deposition rate in the manufacturing of complex structures. In this paper, a three-beam coaxial wire system is studied, with particular attention to the effects of deposition height and laser defocusing on the resulting bead geometry. As the deposition standoff distance changes, so does the workpiece illumination proportion, which describes the ratio of energy going directly into the feedstock wire and into the substrate. Single titanium beads are deposited at varying defocus levels and deposition rates and the bead aspect ratio is measured and analyzed. Over the experimental settings, the defocusing level and deposition rate were found to have a significant effect on the resulting bead aspect ratio. As the defocusing level is increased away from the beam convergence plane, the spot size increases and the deposited track is wider and flatter. Process parameters can be used to tune the deposited material to a desired aspect ratio. In coaxial wire deposition, defocusing provides an adjustment mechanism to the distribution of heat between the wire and substrate and has an important impact on the resulting deposit.

## 1. Introduction

Directed energy deposition is an attractive additive manufacturing (AM) technology that is fit as an alternative to traditional manufacturing methods for the fabrication of complex, unique, and large components and for the repair of existing structures [1]. The directed energy deposition (DED) process combines a continuous material (usually powder or wire metals) feed and energy source (commonly laser, electron beam, or electric arc) to additively manufacture near-net shape components. Compared to classical manufacturing technologies and other AM techniques, DED is more sustainable, creates less waste, and can take less time to fabricate a part [2]. The field is split into two main categories of powder and wire systems. Powder DED systems have the benefits of higher achievable resolution, the ability to easily alloy metal powders, and efficient energy absorption at the expense of lower deposition rates, more difficult material preparation and handling, more waste created, and posing a health hazard to operators who might breathe in metallic particulates [3]. Wire systems, on the other hand, are usually used for applications that require high deposition rates and/or are for relatively large builds. Although wire systems generally have a lower achievable resolution and roughness between layers, they provide benefits of much reduced waste, process efficiency, simplified material preparation, and high material quality with a low defect rate [4].

Coaxial wire-based laser metal deposition is a DED strategy that features wire delivery that is perpendicular to the substrate surface and a laser that is delivered around the wire axis. The process is less sensitive to wire feeding direction when compared to conventional off-axis wire deposition systems [5]. The optics head in a coaxial setup is generally more complex as the laser beam has to be split and aimed at the wire [6,7].

Process parameters that are important to the coaxial deposition process are laser power, wire feed speed, traverse feed rate, laser spot size, laser delivery style, and laser delivery angle [8]. The combination of the process parameters as well as temporal aspects of the deposition determine the quality and defect rate of the printed material. Due to the unique arrangement of the wire and lasers (as shown in Figure 1a), the standoff distance of the process affects the interaction of the laser with the wire. As the deposition standoff distance changes, so does the workpiece illumination proportion (WIP), which describes the ratio of energy going directly into the substrate and into the feedstock wire [9]. If the standoff distance from the substrate is decreased, the lasers will intersect the substrate earlier in their beam path, resulting in a higher WIP and more laser energy being directly input to the substrate. The beam spot pattern and interaction with the wire and substrate are pictured in Figure 2. The adjustments to standoff distance away from the plane where the beams converge is referred to as defocusing. The defocusing of the beams is an important mechanism to determine how much energy goes into heating the wire and the substrate, the size and shape of the laser spot, and, more generally, the distribution of heat at and around the melt pool. Important deposition characteristics like bead shape, residual stresses, defect rate, and adhesion to the substrate surface are determined by this heat distribution.

The study of the effects of defocusing and WIP in coaxial deposition has been generally concentrated on the study of process stability [9,10,11,12,13,14], bead geometry and substrate dilution [12,14], and mechanical properties [14]. Kotar et al. found that, with increasing WIP, the stability process window becomes larger, as indicated by the increase in the range of minimum to maximum allowed laser power for a fixed wire feed rate and traverse feed rate [12]. This work also found that, with all other variables held constant, WIP did not have an impact on bead width and height over the tested ranges but rather only had an impact on the substrate dilution. Conversely, Govekar et al. and Ji et al. claim that the WIP parameter strongly influences the deposited layer geometrical properties, potentially suggesting that large changes to the WIP setting yield meaningful changes to the bead geometry [10,14]. These studies have been limited to the coaxial wire deposition of nickel, steel, and stainless steel. With different cooling and surface tension properties of Ti-6Al-4V (Ti64), the current paper will be novel in its experimentation and simulation of the effects of defocusing in titanium coaxial wire deposition. Additionally, most studies on the effects of WIP in coaxial wire deposition are for setups with an annular laser beam. This work examines a unique three-beam setup. This work also serves to confirm the experimental trends found in the previous literature for commonly used engineering metals. The introduction of a stationary physics-based simulation provides rapid insight to the process that can improve deposited material without having to expend expensive resources, feed stock, substrates, and deposition time.

**Figure 1 materials-17-00889-f001:**
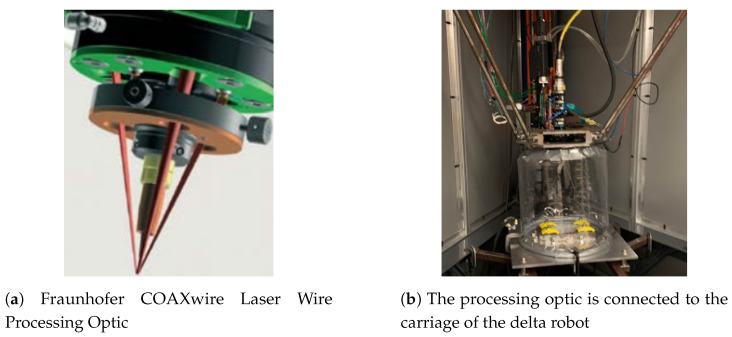
Setup of processing optic integrated with delta robot, laser input, and wire input [15].

**Figure 2 materials-17-00889-f002:**
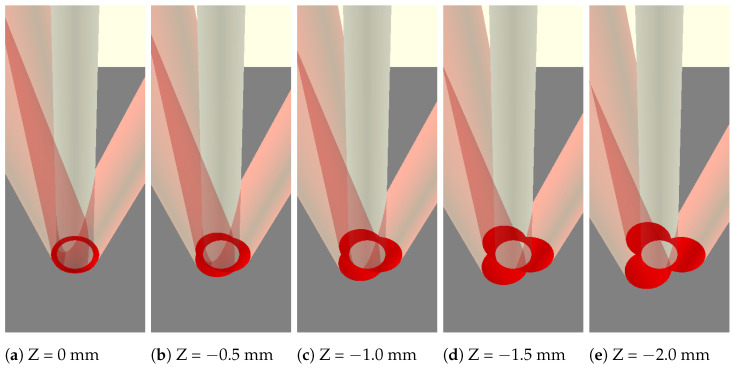
Beam spot patterns at various work plane offsets. The Z position denotes the distance between the beam convergence and work planes. WIP increases with increasing defocusing distance.

The current paper first discusses a description of the experimental setup and design and a simulation of the laser-wire process to study the impact of important process parameters on the resulting bead shape. The experimental results are presented and statistically analyzed and validated. Insights from the simulation are presented and correlated to the experimental results. Finally, further research directions are shared and final conclusions from the study are drawn.

## 2. Materials and Methods

### 2.1. Experimental Setup

The setup includes a laser, a wire feeding system, a three-beam coaxial processing head, and a robot positioning system. The laser is a 4 kW Laserline LDF 4000-30 diode near-infrared (NIR) laser that features a top-hat beam profile. The wire feeding system is a Miller Auto-Continuum 500 welder with an attached bulk and precision feeder. A Miller Auto-Continuum Wire Drive is used for bulk wire feeding and a Dinse DIX FD 200 M feeder used for precise feeding. The feedstock in the system is 1.2 mm diameter Ti64 wire and is printed onto 6 mm thick Ti64 substrates. Both the laser and wire inputs are delivered to the Fraunhofer COAXwire Laser Wire Processing Optic on the same axis. This delivery style is what differentiates this technology from other wire DED systems. The laser is split into three discrete beams that are spaced symmetrically around the wire and have an outer beam aperture angle of 40°. Figure 1a shows the arrangement of the three laser beams and the wire [15]. The coaxial head is the end effector of a three-axis delta robot that is used for positioning the depositor for printing. There is a flexible plastic bag that is connected to the head and the substrate that is filled with argon to reduce oxidation in deposits. Figure 1b shows the deposition cell. In support of the experimental research at Missouri University of Science and Technology (MST), GKN Aerospace provided the laser, coaxial wire deposition head, and titanium wire and substrates, as well as relevant technical support for the deposition cell.

Since the Fraunhofer optics head has three laser beams that are delivered coaxially to the feedstock wire, there are a few features of the process that are unique. The wire is heated at three points around its diameter, making for more distributed heating when compared to other wire and laser delivery strategies. Furthermore, by changing the standoff distance of the deposition head to the substrate surface, the interaction of the laser beams with the wire can be adjusted. There is a standoff position where the laser beams converge to make a concentrated laser spot. Any deviation from this spot diverges the beams and distributes the laser energy over more area. Figure 2 shows multiple different standoff distances adjustments and the resulting beam spot pattern on the wire and substrate. At the standoff distance that corresponds with laser beam convergence, the proportion of laser energy going straight into the wire is high, pictured in Figure 2a. The WIP metric quantitatively describes this idea, specifying the proportion of laser energy that is delivered directly to the workpiece or substrate. The remainder of the laser energy is delivered directly to the wire. The work plane can be offset away from the beam convergence plane to change the beam spot pattern and resulting WIP. With increasing defocusing distance comes increased WIP as more energy is directly input to the substrate. This has the effect of enabling adjustment over the effective spot size of the combined laser beams, the amount of energy going to remelting workpiece material, and the amount of energy going to melting the feedstock wire. In this study, WIP will be studied for its effect on the deposited bead geometry.

### 2.2. Experimental Design

To study the effect of different deposition parameters on the bead geometry, single-bead tracks were deposited and compared. To ensure the varied experimental runs were comparable, certain parameters were held constant in the design of the experiment. The track length and amount of material in each bead were constant by fixing the ratio of traverse feed rate to wire feed speed (WFS). By having a constant amount of material in each bead, the cross-sectional area of each bead should be the same, making for a good comparison between beads of the same size but of different shapes. Hereafter, changes to the traverse feed rate will imply a proportional change to the WFS and deposition rate. For each treatment combination (TC) to result in a stable deposit, the laser power (LP) for each experimental run had to be adjusted. After running a pilot study to examine the stable process windows for each treatment combination, the stable experimental laser power for each run was found to follow a pattern described by Equation (Equation 1). The physical interpretation of this equation is a constant amount of energy per unit length (J/mm) into the wire. This equation does not strictly define what a stable process is. Instead, it helped to define an appropriate and consistently calculated LP for the study that was not arbitrarily chosen. It was defined as part of the current work to achieve this purpose.
(1)LP∗(1−WIP)/Feed=Constant

The experiment was designed with two factors with three levels each. The factors and associated levels are traverse feed rate (hereafter referred to as feed) at 5.927 mm/s, 7.197 mm/s, and 8.467 mm/s and WIP at 25.20%, 40.00%, and 55.00%. Figure 3 shows the spot patterns at each of the experimental settings of WIP. The lowest WIP setting of 25.20% represents the case in which the work plane is set at the beam convergence plane and the WIP is minimized for the combination of laser spot size, laser delivery strategy, and wire diameter. Table 1 shows the nine treatment combinations and their associated experimental deposition parameters. The run order for the nine treatment combinations was randomized and run in sequence all on one substrate. This process was replicated once for a total of 18 experimental runs across the two blocks of different substrates. Figure 3 shows the run order for each of the blocks.

### 2.3. Experimental Procedure

Nine beads were deposited on each plate according to the randomized run order and specified process parameters to make up the 18 total experimental runs. Each bead has a length of 76.2 mm, and they were separated by 10.5 mm. Pictures of three sample beads are shown in Figure 4. To measure the profile of the deposits, each plate was scanned using a Revopoint MINI 3D Scanner with a precision of 0.02 mm. Prior to scanning, the deposits were sprayed with a thin layer of AESUB Orange Scanning Spray to make the surface non-reflective. The scan produced an STL file of the nine beads and the substrate surface. Figure 5a shows the height profile of the scanned deposits at the midpoint of the length of the beads. From the raw midpoint scan data, it can be seen that there is significant warping in the substrate due to induced residual stresses from the deposition process. To properly measure the bead geometry, the curvature of the substrate must be removed. Figure 5b shows the raw data decomposed into the red substrate lines and the blue peaks that represent the deposited beads.

Using the decomposed scan data, the substrate was flattened to its original state prior to deposition and the peaks were flattened and reattached to the substrate, shown in Figure 6a. From the conditioned profile data, the maximum value of each peak was found to measure the height of each bead. The width of the beads was measured as the bead width 0.05 mm above the flattened substrate surface. Figure 6b graphically shows the width and height measurements.

To quantitatively capture the bead shape, the aspect ratio (AR) was calculated for each deposited bead. The aspect ratio is the ratio of the bead width to the bead height. Since each bead has the same amount of deposited material and the same cross-sectional area, AR encodes differences in the shape between the beads.

### 2.4. Simulation Description

The simulation model that was used to better understand the coaxial deposition process was developed at Product Innovation and Engineering LLC (PINE) in conjunction with MST. This model specifically targets metal AM using the DED method; however, it is extensible to nearly any AM process. The model, described in more detail in [16], is a material-centric physics-based simulation with the express goal of being as efficient as possible while still generating results that provide guidance when making path planning decisions by providing a thermal history for a given build. This model has been shown to provide accurate results for Ti-64 using the material properties detailed in [17].

The simulations for this body of work all had the same setup, which was to have a wire contacting the substrate at the center of a block of material. Then, three lasers were projected such that they met at the central axis of the wire, just as in the physical setup shown in Figure 1a. The location of the intersection was moved, just as in the experimentation, to evaluate the effect of the defocusing of the lasers on the temperature profile developed. The laser was pulsed for 250 ms and the entire simulation was allowed to cool in order to show the effect of the heating from that WIP value. Sample images from each of the different WIP values used can be seen in Figure 7. The parameters used for this simulation setup can be found in Table 2 and the material properties for Ti-64 can be found in [17]. These models were all run on a custom Linux PC with a Ryzen 9 3900X CPU and a Radeon RX 6900XT GPU. These models took on average 20 h to complete the simulation.

## 3. Results

### 3.1. Experimental Results

To reduce the impact of any profile variations along the length of a bead, the AR was measured at 40%, 50%, and 60% of the bead length. These three measurements were then averaged for a single AR output value for each of the 18 experimental runs. Table 3 shows the run order of the treatment combinations for each of the experimental blocks (plates) and the output mean ARs. Figure 8 shows the midpoint scans for all the beads. Each treatment combination is labeled and arranged to correspond to the feed rate and WIP settings.

### 3.2. Simulation Results

To analyze the simulation results, several plots were created. The first of these was created to inspect the temperature along the center axis of the wire. For each time step saved to disk, the temperatures at six different z values were selected. The first was at the intersection of the wire and the substrate (z = 0.0 mm), three were taken within the wire (z = 9.0 mm, z = 6.0 mm, and z = 3.0 mm), and two were measured in the substrate (z = −3.0 mm and z = −6.0 mm). These were plotted in Figure 9. This method showed that the maximum temperature of the central axis of the wire always occurred where contact with the substrate occurred, z = 0.0 mm, at 2074 C, 2384 C, and 2255 C for a WIP of 25.2%, 40%, and 55% respectively. This value was always reached during the laser irradiation for the first 0.25 seconds of the simulation. The heat distribution was very dependent on the WIP value, as can be seen in the distribution of the lines, where lower WIP values had temperature profiles that were much more widely distributed and higher WIP values had a much tighter thermal distribution. This follows the expected trend that a lower WIP value will introduce more energy directly into the wire and a higher WIP value will introduce more energy into the substrate. In the substrate, there is more thermal mass for the heat to be quickly distributed, resulting in a much tighter thermal distribution. Meanwhile, when heat is introduced into the wire with a much smaller thermal mass, it cannot dissipate as much and will create a much greater thermal distribution.

## 4. Discussion

### 4.1. Experimental Discussion

The statistical software JMP and the Python library statsmodels were used to analyze the experiment data. The experimental variables of feed rate and WIP were normalized over their ranges between 0 and 1. To statistically determine the effects of feed rate and WIP on the resulting aspect ratio, a one-way analysis of variance of the experiment was conducted as shown in Table 4. The null hypothesis of this analysis is that at least two of the treatment combinations have statistically equal means and the alternative hypothesis is that all treatment combination means are unequal. With a *p*-value less than 0.0001, we reject the null hypothesis and know that the treatment combination means are unequal at a 0.05 significance level. Next, we look to the effects tests in Table 5 to see which factors have a significant impact on the AR of a deposited bead. At a 0.05 significance level, the interactions of the two factors and the blocking factor are both insignificant. The main effects of WIP and feed rate as well as the square of feed rate are all significant at the specified significance level. The squares of the factors aim to see if there is significant curvature in the model. If the output is a linear function of the tested input variable, its square term will be insignificant. This can be seen graphically and is detailed more below in Figure 10.

Using only the significant factors, a new model was created for which the parameter estimates for the normalized variables are shown in Table 6. The model was generated using the statsmodel Python library and is a multiple linear ordinary least squares regression. The effects plots for this model were generated and are shown in Figure 10. To show the general trend of each of the experimental factors, the model fit at 0.5 of the normalized variables was plotted along with all the measured aspect ratio results. Each of the plots shows a general trend upwards in aspect ratio with increasing either factor, although the feed effect has some curvature. Over the tested ranges, changes to the WIP had a larger impact on the output aspect ratio. This can be seen in Figure 8, where the changes in the WIP setting had a greater effect on the resulting bead shape compared to the changes in the feed rate setting. These estimates yield Equation (Equation 2), which predicts the bead AR for a given feed rate in mm/s and a WIP in % over the respective original parameter ranges. The R-squared value for the model is 0.926, which indicates a good fit between the model and the experimental data.
(2)$$\begin{array}{c} 5.785+0.019\times \mathrm{WIP}-1.457\times \mathrm{Feed}+0.109\times {\mathrm{Feed}}^{2}={\mathrm{AR}}_{\mathrm{Model}}\\ R^{2}=0.926 \end{array}$$

To test the performance of the model with previously untested treatment combinations, two model validation runs were deposited and their bead geometries were measured. Table 7 shows the deposition process parameters for each run and the resulting mean measured AR and model-predicted AR. For each validation run, the model prediction has about 4% error from the experimentally measured value of AR. Figure 11 shows midpoint scans of the validation deposits.

Using the information from the study, some main conclusions can be drawn. First, the data from the experiments show how the deposition factors impact the bead geometry of Ti64 coaxial wire deposits. Different wire delivery strategies, wire materials, and wire diameters could result in varying results of the effects of WIP and feed rate on bead geometry. For this setup, the data from these experiments can be used to inform deposition settings that will result in the desired bead geometry. Any of these parameter sets can be used to make stable single tracks, but, as more complex builds are made, the bead shape will be an important factor in determining the amount, size, and type of defects present in the build. It is also worth emphasizing that, over the tested ranges, there was no significant interaction between the experimental factors with respect to bead aspect ratio. This allows for their effects to be studied independent of each other.

It is clear that, as the WIP increases, the bead becomes wider and flatter. This can be attributed to a larger melted area. A wider and flatter bead is generally desirable as they are easier to stack in structures that are multiple beads wide and tall. This benefit is bounded by the need to have enough laser energy going into the incoming wire to melt it and ensure a stable process. The amount of energy delivered to the wire can be adjusted by changing laser power, but this also changes the energy going to the workpiece. This balance of the correct amount of energy in the right places is important to ensure that the material is being melted without excess energy regarding vaporizing material. The traverse feed rate had a more complicated effect on the bead aspect ratio. There was not much of a difference in the aspect ratio produced by the two lower settings for feed rate. At the higher feed rate, the bead aspect ratio was discernibly higher. This suggests that, at higher feed rates, the melt pool solidifies quickly enough to not allow the surface tension of the liquid melt pool to pull the bead into a more narrow track. Many trade-offs exist between process parameters such as available laser power, deposition rate, wire feed speed, and WIP that must be considered to achieve a particular desired result.

The current experimental study is limited in its scope of only evaluating the bead aspect ratio and not considering other important characteristics like penetration depth, deposit to substrate bond strength, microstructure, mechanical properties, and deposition stability as a function of WIP. Capturing and understanding the bead shape of a parameter set is a critical first step in building towards fully dense AM builds. The study also only focused on the effects of deposition rate and WIP on the process, but there are certainly more variables that impact the characteristics of a build. Varying laser delivery strategies, wire materials, and wire diameters should be studied to see if the researched trends hold for the tested variables and other process parameters.

To continue the study of coaxial wire deposition and the effects of WIP on the process, the research team will next study the directional effects of the three-beam process. Since the lasers heat the wire from multiple sides, the process is generally understood to have directional independence. However, with three discrete beams, there can be different heat distributions with respect to the travel direction. For example, the deposition direction can be directly towards one of the beams, directly between two of the beams, or any position in between. In the various cases, the leading and trailing sides of the melt pool are going to be heated differently. This effect might be amplified as the beams diverge with increasing WIP. In the longer term, the research group hopes to make multi-bead wide and multi-bead tall structures that are free of large and critical defects when depositing in any direction.

### 4.2. Simulation Discussion

To compare the model and use it as a predictor of the experimental results, the temperature profile for each of the WIP values was plotted for z = −3.0 mm, z = 0.0 mm, and z = 3.0 mm in Figure 12a–c, respectively. This showed that, at the interface of the wire and the substrate, z = 0.0 mm, there was limited difference for the duration of time when the laser was on. However, as the build cooled, there was a visible difference due to the distribution of the heat in the wire vs the substrate. This trend of differing distributions can also be seen at z = −3.0 mm and z = 3.0 mm, where a higher WIP will put more energy into the substrate and a lower WIP will put more energy into the wire.

This trend carried over to additional analyses that were conducted, the first of which measured the height of the melted region within the wire for the various WIP values, and another mode of analysis plotted the radius circle that circumscribed the melted region on top of the substrate; these are shown in Figure 13 and Figure 14, respectively. From the radius of the circle that circumscribed the melt pool plot, in Figure 13, it can be seen that a higher WIP correlated to a larger radius and a lower value correlated to a smaller radius. This was expected due to the increased amount of laser energy that is directly applied to the substrate as the WIP is increased. The plot of melt pool radius closely matches the trends found in the experimental study. Higher WIPs resulted in increased simulated melt pool radius and increased experimental bead widths and aspect ratios. On the contrary, a higher WIP correlated to a smaller section of the wire being melted. This again was expected due to the distribution of the laser energy being located on the wire and substrate.

These plots can be used in future work to develop a robust processing window for the DED process. With some experimental support, it would be possible to determine a maximum allowable region of the wire that can melt before balling occurs. This mode of failure can be detrimental to the print quality and should be avoided. Using this model, it would be possible to map out the allowable laser powers such that balling does not occur. This would need to take into account the wire feed speed as well since more melting would be possible if the wire was inserted into the melt pool faster. Additionally, the radius of the circle that circumscribes the melt pool can be used as a predictor of the wire’s ability to wet to the substrate. If a large enough melt pool is not created, there is not sufficient area for the wire to wet and properly adhere, which will lead to a poor overall build quality.

The last method of analyzing the data took into account these last two results and calculated the percentage of the melt pool that was located in the substrate vs in the wire. This is shown in Figure 15. This plot showed that the WIP value predicted the amount of the melt pool that would be in the substrate; indeed, the smaller the WIP, the less the melt pool would be located in the substrate. This last method of comparing the distribution of the melt pool location has been the one that appears to be the best predictor of the aspect ratio of the completed build. This is due to its accounting for the amount of wire that is melted and can be placed onto the substrate along with the amount of substrate that is melted and ready to receive the new material. In cases of low WIP, most of the melted material is in the wire with a small amount in the substrate, making for a narrow deposited track. With higher WIPs, there is more melted material in the substrate, which allows the melted wire material to more easily spread out and make a wider bead. These trends are supported by the experimental study. This style of simulation with consideration of additional process parameters can be used as a predictor of deposition stability, bead geometry, and substrate dilution.

This simulation setup is primarily limited by not moving the laser and wire and thus not capturing the effect of feed rate on the process. For this study, the movement of the melt pool was excluded to see if a simple, quicker, and less computationally expensive method could inform the coaxial deposition process. In future studies, the simulation will include movement of the melt pool. Additionally, the wire in the simulation was not being fed into the process during the simulation. The introduction of relatively cool wire into the process can provide more material for heat to distribute to. With the compensation of wire feed and traverse feed motion, the simulation can be used to fully simulate a deposited track.

## 5. Conclusions

The coaxial wire laser DED technique is an attractive option for the fabrication of large metal components because of its high achievable deposition rates. At the expense of the high rate are increased minimum feature size and resolution. The technology has unique features and capabilities that need to be further studied to ensure that the process is robust, repeatable, and capable of meeting the requirements asked of various applications. Since the process is most cost-effective for large parts, the best applications of the technology are in the aerospace industry and as a replacement to large castings, dies, and molds. Although the present study was focused on smaller single-bead structures, the technology can be easily scaled to larger deposition tracks and larger builds.

This paper provided an experimental and physics-based simulation analysis of coaxial wire-based laser metal deposition. The unique arrangement of the laser beams and wire feedstock allows for the adjustment of the heat distribution in the wire and substrate through changes to the standoff height. Through making changes to the important process parameters of WIP and deposition rate, changes in the resulting bead geometry were analyzed. The research yields the following conclusions:The workpiece illumination proportion (WIP), as set by standoff distance, has an important role in determining the heat distribution, melt pool size, amount of melted material, and bead height and width.For a given bead size, the deposition rate has a significant impact on the resulting bead shape. At the highest experimental settings of traverse feed rate and wire feed speed, the bead aspect ratio was noticeably increased as melt pool solidification kinetics are less dominated by the surface tension effect.A statistical regression model that predicts bead aspect ratio was presented and validated for the experimental study of WIP and deposition rate in coaxial wire-based laser metal deposition. Although the specific values in the model might only apply to the specific sizes, beam arrangement, and materials in the experiment, the general trends can provide insight into phenomena of other laser additive setups.The physics-based simulation provided a framework for the analysis of the coaxial wire-based laser metal deposition process. The calculation and analysis of the radius of the melt pool, the amount and location of melted wire, and the temperature of regions of interest can help to have a better understanding of important deposition characteristics like stability and bead geometry.

## Figures and Tables

**Figure 3 materials-17-00889-f003:**
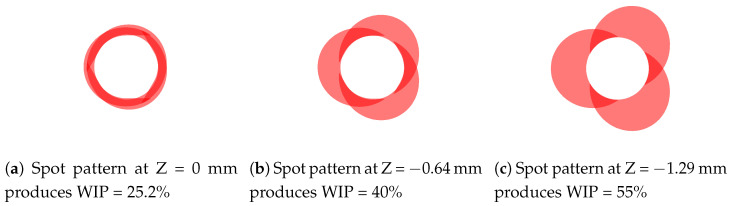
Variations in the WIP value for the experiment were achieved by placing the beam convergence position below the work plane, resulting in the beam spot patterns indicated in (**a**–**c**).

**Figure 4 materials-17-00889-f004:**
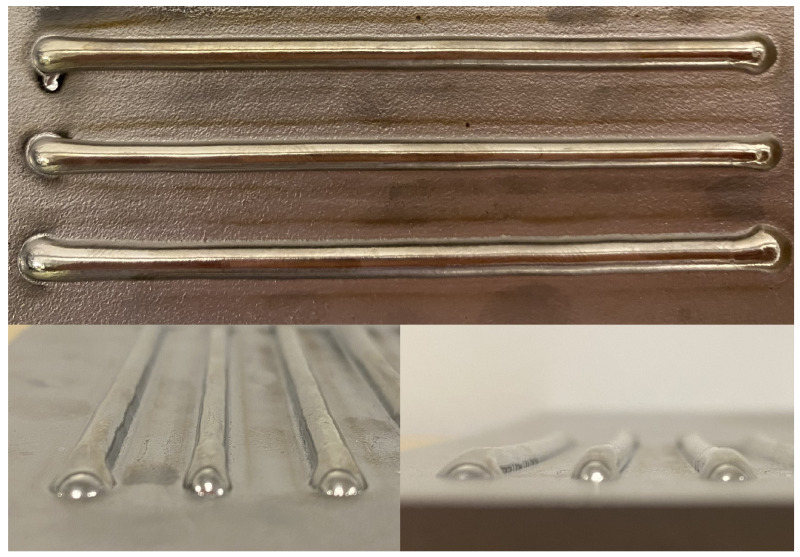
Three sample deposited beads.

**Figure 5 materials-17-00889-f005:**
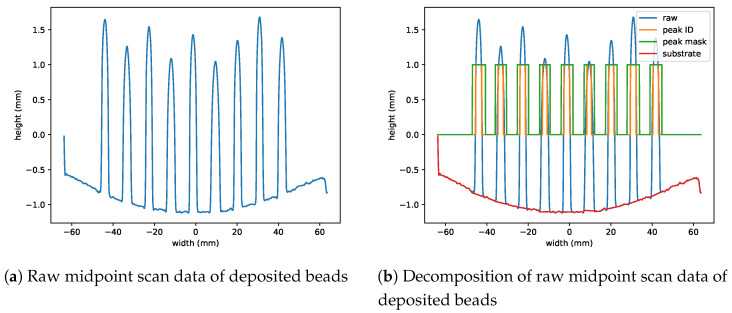
Midpoint scan data with substrate warping. Because of the shrinkage of the liquid material as it solidifies, there is curvature in the substrate after it is unclamped from the work holding. Using the decomposition, the curvature can be eliminated.

**Figure 6 materials-17-00889-f006:**
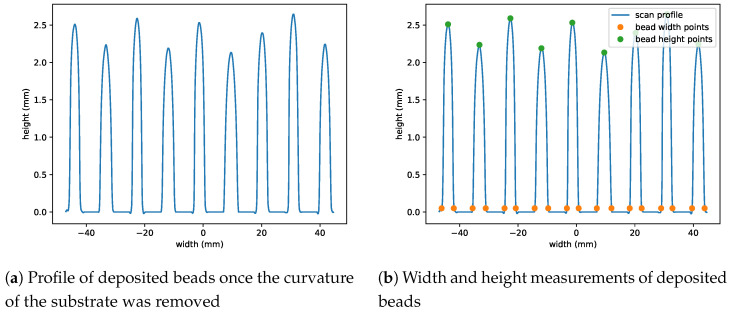
Midpoint scan data with substrate warping removed. From the newly constructed data, the width and height of each bead were measured.

**Figure 7 materials-17-00889-f007:**
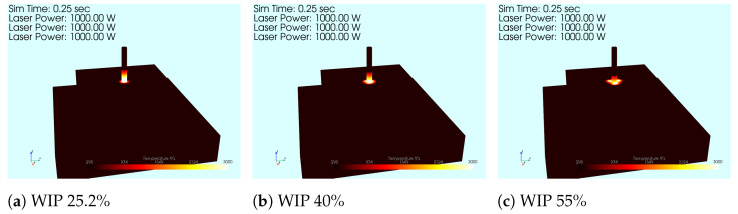
Simulation images from various WIP values. At a lower WIP, more of the wire is heated to an elevated temperature. As the WIP increases, the heat is spread out over more area.

**Figure 8 materials-17-00889-f008:**
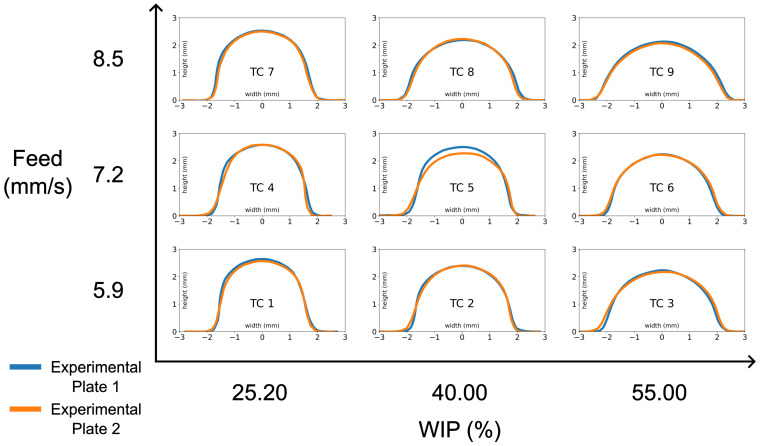
Cross-sectional profile at the midpoint of the beads for all nine tested treatment combinations of varying feed rate and WIP. The two plotted bead profiles on each graph represent the two experimental runs per treatment combination.

**Figure 9 materials-17-00889-f009:**
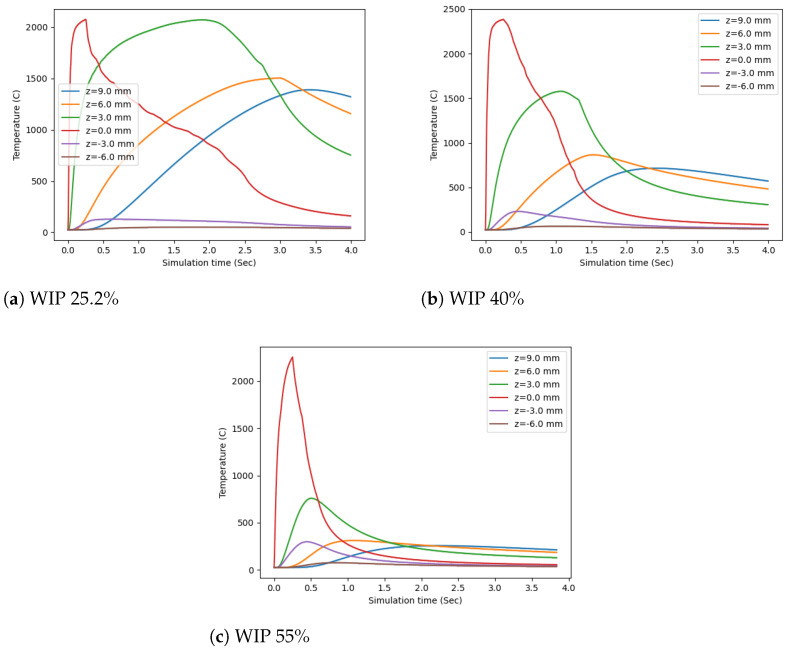
Temperature at the center of wire vs simulation time.

**Figure 10 materials-17-00889-f010:**
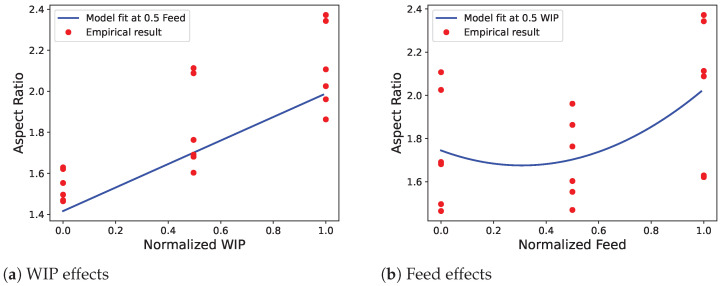
Effects plots of the tested variables.

**Figure 11 materials-17-00889-f011:**
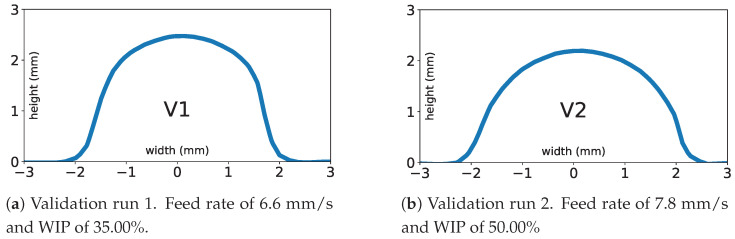
Midpoint scan data of model validation runs.

**Figure 12 materials-17-00889-f012:**
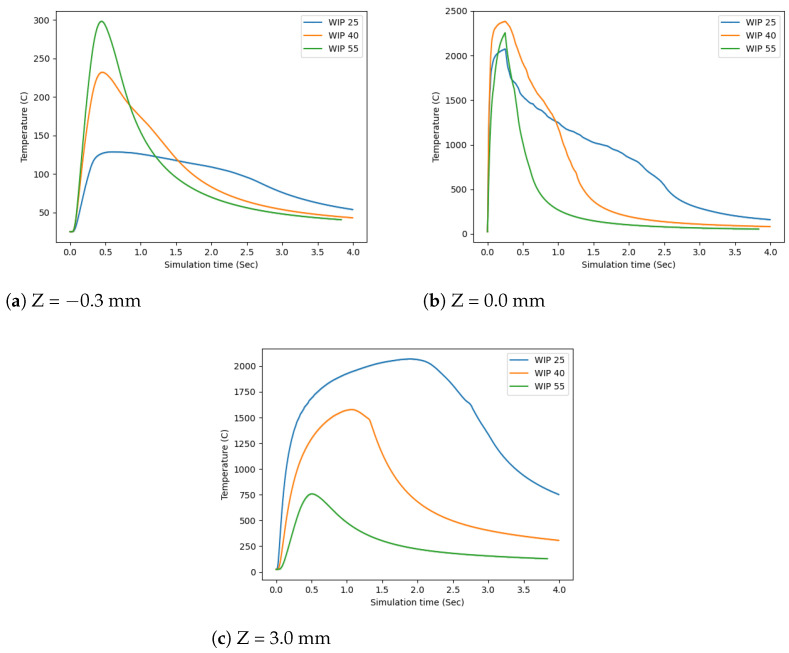
Comparison of the temperature of the wire center at particular Z levels vs. time.

**Figure 13 materials-17-00889-f013:**
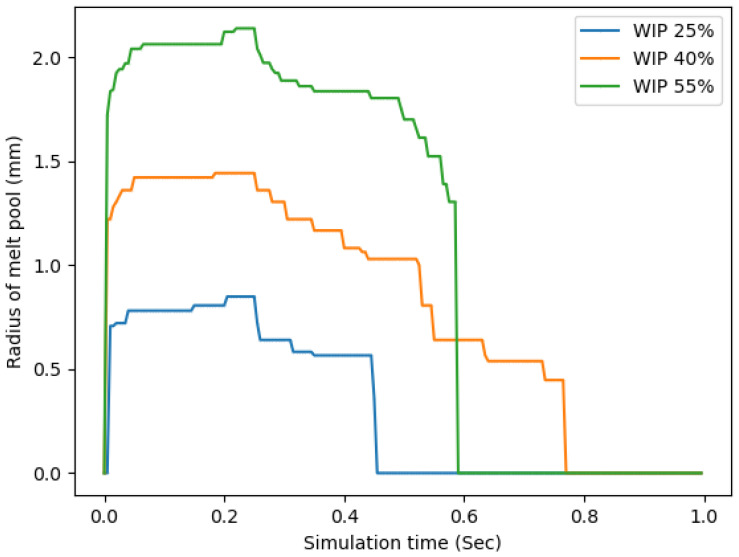
Radius of the circle that circumscribes the melt pool vs simulation time.

**Figure 14 materials-17-00889-f014:**
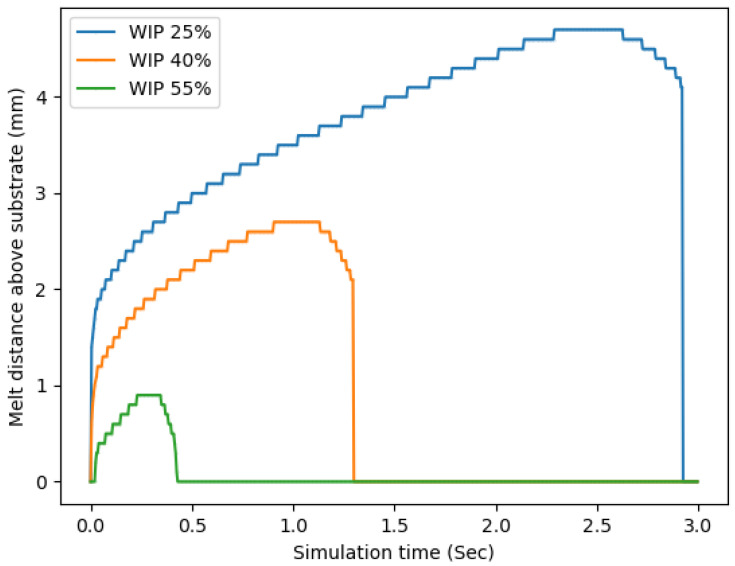
Height of the melted region of the wire vs simulation time.

**Figure 15 materials-17-00889-f015:**
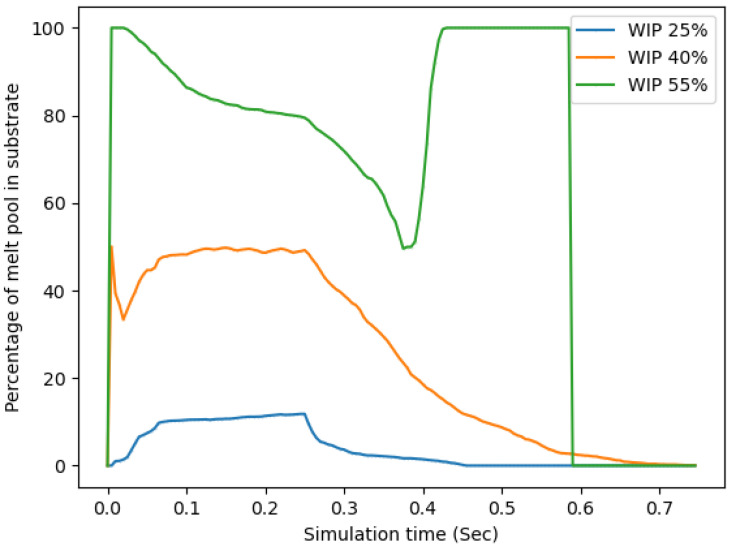
Percentage of the melt pool in the substrate vs simulation time.

**Table 1 materials-17-00889-t001:** Experimental process parameters.

Treatment Combination	Feed (mm/s)	WFS (mm/s)	WIP (%)	LP (W)
1	5.9	38.3	25.2%	1029
2	5.9	38.3	40.0%	1283
3	5.9	38.3	55.0%	1711
4	7.2	46.6	25.2%	1250
5	7.2	46.6	40.0%	1558
6	7.2	46.6	55.0%	2078
7	8.5	54.8	25.2%	1471
8	8.5	54.8	40.0%	1833
9	8.5	54.8	55.0%	2444

**Table 2 materials-17-00889-t002:** Simulation setup parameters.

Parameter	Value
Resolution	10 μm
Laser Power (per laser)	1000 W
Laser Profile	Top Hat
Laser Diameter (per laser)	1.5 mm
Laser Pulse Duration	250 ms
Rotational angle separating lasers	120°
Angle between laser path and wire axis	20°
Substrate dimensions	25 mm × 25 mm × 8 mm
Wire Diameter	1.2 mm

**Table 3 materials-17-00889-t003:** Experimental run order and output aspect ratios.

Block 1					Block 2				
TC	AR 40%	AR 50%	AR 60%	Mean AR	TC	AR 40%	AR 50%	AR 60%	Mean AR
5	1.582	1.654	1.572	1.603	7	1.638	1.581	1.643	1.621
3	2.156	2.000	1.919	2.025	6	1.896	2.039	1.948	1.961
4	1.585	1.483	1.590	1.553	1	1.482	1.528	1.477	1.496
8	2.089	2.118	2.057	2.088	9	2.250	2.411	2.368	2.343
7	1.577	1.589	1.721	1.629	8	2.247	1.967	2.125	2.113
9	2.402	2.349	2.364	2.372	3	2.005	2.216	2.099	2.107
2	1.638	1.711	1.695	1.681	2	1.709	1.721	1.643	1.691
1	1.430	1.465	1.497	1.464	4	1.445	1.475	1.488	1.469
6	1.863	1.915	2.014	1.931	5	1.762	1.850	1.676	1.763

**Table 4 materials-17-00889-t004:** Analysis of variance.

Source	Degrees of Freedom	Sum of Squares	Mean Square	F-Ratio
Model	6	1.3988	0.2331	31.6574
Error	11	0.0810	0.0074	* **p** * **-Value**
C. Total	17	1.4798		<0.0001

**Table 5 materials-17-00889-t005:** Effect tests.

Source	Degrees of Freedom	Sum of Squares	F-Ratio	*p*-Value
Feed	1	0.2414	32.7794	0.0001
WIP	1	1.0246	139.1303	<0.0001
Feed × WIP	1	0.0106	1.4395	0.2554
Feed × Feed	1	0.1190	16.1622	0.0020
WIP × WIP	1	0.0001	0.0180	0.8956
Block	1	0.0026	0.3585	0.5615

**Table 6 materials-17-00889-t006:** Parameter estimates.

Term	Estimate	Std Error	t Ratio	*p*-Value
Intercept	1.4581	0.044	33.213	<0.001
WIP	0.5732	0.051	11.299	<0.001
Feed	−0.4517	0.183	−2.469	0.027
Feed × Feed	0.7353	0.176	4.184	0.001

**Table 7 materials-17-00889-t007:** Validation runs of experimental fit model.

TC	Feed (mm/s)	WFS (mm/s)	WIP (%)	LP (W)	AR 40%	AR 50%	AR 60%	Mean AR	Model AR
V1	6.6	42.5	35.00%	1312	1.633	1.686	1.623	1.647	1.581
V2	7.8	50.7	50.00%	2035	2.129	2.068	2.080	2.092	2.002

## Data Availability

The original contributions presented in the study are included in the article. Further inquiries can be directed to the corresponding author.

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
