# Peer review of "Effects of Laser Defocusing on Bead Geometry in Coaxial Titanium Wire-Based Laser Metal Deposition"

_materials, 2024, doi:10.3390/ma17040889_

Round 1
Reviewer 1 Report
Comments and Suggestions for Authors
This paper by Mathenia et al. presented a study of the effects of laser defocusing on the bead geometry in the direct energy deposition (DED) process. Both experimental tests and simulations were conducted, and the results were reported in the paper. Overall, the paper has good quality, and the content of this work may be interesting to the readers of Materials. The reviewer has the following comments:
- It appears that three laser beams are included in the DED setup. The term “coaxial” may be confusing since “co” describes two items. It is advised that the coaxial term be changed.
- Authors should consider adding a photograph of the printed samples, such that the “bead geometry” is clearer to readers.
- In Figure 7, it is unclear which color represents the experiment tests and the simulations. All the plots should include legends. The same question goes with Figure 5. The meaning of the green curve and the red dots should be clarified in the legend.
- The caption of the figures should be placed together instead of separately. (Fig 1, 3, 4, 5, 9, 10, 12).
- The method of the statistical analysis should be specified. It appears that the authors used Analysis of Variance (ANOVA). Then what are the null hypothesis and alternative hypothesis? Is the one-way ANOVA method used? Or is two-way used?
- In Table 5, what do the square of the feed and the square of the WIP mean? And if the Feed*Feed and the WIP*WIP are both significant, what does it indicate? The authors should add more discussion. What does the column of “Prob>F” mean? Are these numbers in Prob>F column the P value?
- It seems that the regression analysis is also used. Again, the methodology should be specified.
- Many abbreviations are not listed, and the authors should carefully check on them.
The quality of the English language is acceptable.
Reviewer 2 Report
Comments and Suggestions for Authors
In this study, Remy et al. discussed the effects of laser defocusing on bead geometry in coaxial titanium wire-based laser metal deposition based on three-beam coaxial wire system. They found that the spot size increased and the deposited track is wider and flatter as the defocusing level is increased away from the beam convergence plane. However, several concerns need to be addressed before it is acceptable.
1) The authors need to further highlight the innovation of this study by comparing to the previous literatures results in the Introduction section.
2) In the Figure 7, what is the meaning of both orange and blue color curves? There was no information in this figure.
3) The range of stimulation time is 0.0-4.0 sec in Figure 11, but the range of stimulation time is 0-1 sec (Figure 12), 0-3 sec (Figure 13), and 0-14 sec (Figure 14). Why did not the stimulation time in these figures stay in the same ranges?
4) The author has addressed various parameters for the coaxial deposition process. It’s better to give an outlook, including advantages and disadvantages, promising applications, etc.
Reviewer 3 Report
Comments and Suggestions for Authors
Line 21. The comparison is made with classical manufacturing processes or other AM technologies? Please clarify.
Figure 3. Are the values transformed from inches to mm? If so, please mention also the inches in brackets (0.025 and 0.05 in), if this is correct.
The nine bed depositions from Figure 4 and Figure 5 are made with which spot pattern/WIP from Figure 3? It would be helpful to have the same comparison for each WIP. L.E.: This was mentioned in Figure 6, as a simulation process and later during the results.
I suggest to overlap the image in Figure 10. Although the difference is noticeable, it would be great to have them on the same image to a better comparison.
In the conclusion section, the authors mention a comparison between experimental and simulation. However, the paper mostly refers to the process simulation at different parameters. The only mention for the experimental is discussed in Figure 4 and Figure 5, where the deposits were scanned. An image of the deposits must be presented in the paper, with top and side views, to have a visual aid of the deposits.
Also, a visual comparison between the melt pool vs. simulated one would be a great plus. The simulation itself has to be correlated to the experimental results.
At least the physical properties (metallography, density, porosity) could have been addressed, for at least the best process parameters found. It is understood that further analyses are needed, and also physical and mechanical properties of the material should be addressed in relation to the described process parameters.
Reviewer 4 Report
Comments and Suggestions for Authors
The present paper provides an experimental and physics-based simulation analysis of coaxial wire-based laser metal deposition. Further, the three-beam coaxial wire system with particular attention to the effects of deposition height and laser defocusing on the resulting bead geometry is analyzed. This study represents an interesting phenomenon in additive manufacturing technology and the paper is well-structured and well-written. However, this reviewer writes some minor comments and suggestions to revise the paper for wider readability and interest to the researchers.
1. The last paragraph of section 1 should be revised by discussing what is done section-wise in the context of the present manuscript.
2. Equation (1) should be supported by a reference.
3. Figure 1 (a,b), Figure 2, Figure 4 (a,b): Captions need to be reduced, and the labeling of Figure 1 (a,b) should be reduced.
4. The computational cost of the numerical simulations should be discussed.
5. Conclusion: The limitations of the present analysis and its future direction of research should be discussed clearly.
Comments on the Quality of English LanguageMinor editing of English is needed.
